# The Impact of Biochar Used in Repairs to Historical Buildings on Public Health

**DOI:** 10.3390/ijerph192012996

**Published:** 2022-10-11

**Authors:** Daniel Tokarski, Irena Ickiewicz, Wioletta Żukiewicz-Sobczak, Paweł Woliński

**Affiliations:** 1Faculty of Economics and Sociology, University of Lodz, 90-419 Lodz, Poland; 2Faculty of Civil Engineering and Environmental Sciences, Bialystok University of Technology, 15-351 Bialystok, Poland; 3Department of Food and Nutrition, Calisia University, 62-800 Kalisz, Poland; 4Faculty of Technical Sciences, Collegium Mazovia Innovative School, University in Siedlce, 08-110 Siedlce, Poland

**Keywords:** biochar, historical building, eco-innovation, building mycology, public health

## Abstract

The subject matter of this manuscript concerns the analysis and identification of microorganisms that pose a threat to human health and, in particular, mold fungi occurring in historical buildings. Surfaces infected by fungal spores pose a threat to the structure and the health of both visitors to historical buildings and professionals working in them. Research was undertaken to fill in the defects in building partitions with a supplementary layer of biochar in order to eliminate, or partially reduce, the possibility of contamination with and development of harmful mold fungi. In the designed cement mixture, biochar was used as a filler, the task of which was to eliminate the causes that lead to the development of harmful mold fungi. Microbiological analyses of the surface of walls and air in selected buildings were carried out before and after the application of supplementary biochar layers. The inhibitory properties of the material used against the presence and growth of mold fungi were observed. The average number of microorganisms isolated on the tested partitions decreased by between 70 and 100%. As a consequence, the use of this material significantly influenced the air quality of the rooms, which is important for protecting the health of people at work, as well as those visiting historical buildings.

## 1. Introduction

Exposure to mold fungi is largely related to the environments in which humans work and live [1]. Biological factors within living and working environments are suspended in bioaerosols [2]. Research by many authors has proved that exposure to the microclimate of rooms contaminated with mold spores may cause allergic diseases of the respiratory system. Moreover, it has been known for years that mold fungi play a pathogenic role in the allergization of the human body. A relatively common exposure symptom among those at work is, among others, hay fever or bronchial asthma [3,4,5].

Analysis of the literature on health hazards caused by harmful mycological factors proves that people are exposed to mycological etiological factors not only in the work environment, but also in their apartments or homes [6,7,8,9,10,11].

As a result of many studies and analyses, a relationship has been noted between the deterioration of health among people staying in public and private spaces (all locations which constitute a closed building space) and health ailments within the exposed population. According to reference data, health disorders caused by exposure to mold fungi in apartments are collectively known as “sealed building syndrome”, and it is synonymous with sick building syndrome (SBS) and chronic fatigue syndrome (CFS). Exposure of the body to an “excessively polluted indoor environment” is known as building-related illness (BRI). The described ailments are closely related to the subjective feelings of exposed people. Unfortunately, the exposure pertains not only to old and historical buildings, but also to newly built houses and apartments. The observed symptoms of SBS include fatigue, nausea, headaches and dizziness, irritability, and decreased attention span, followed by irritation of the mucous membranes of the eyes, nose, and throat, and reddening of the skin. There are many factors that can cause such ailments, one of which being toxic volatile metabolites. These are also known as “volatile organic compounds” produced by molds (VOCs). Apart from VOCs, an important factor responsible for the occurrence of the so-called sick building syndrome is (1 → 3) β-glucan. Particular attention should be paid to the fact that ailments of this type, if considered in terms of occupational medicine and allergology, should be referred to as an occupational allergy to mold fungi. Unfortunately, the research into such chronic diseases is still at a relatively low level, despite the fact that many types and species of fungi are already classified as biologically harmful agents present in work environments. Etiological factors in inhalation-related allergies include the spores of microscopic fungi. They are referred to as aeroallergens, which are a part of the broadly understood bioaerosols. According to many researchers, from a range of over 120,000 species of fungi, about 80 species can be associated with respiratory allergies. The microscopic size determines the penetration of the inhaled spores into the respiratory tract, which may result in the development of allergic inflammation affecting both the upper and lower respiratory tract [12,13,14].

Mold spores, after being suspended in the air, typically settle on all room elements and, under favorable conditions, can also initiate their development cycle [15]. Molds produce significant amounts of spores which can be transported over a distance of thousands of kilometers as air and dust pollutants [16]. Mold fungi are cosmopolitan organisms found in all climatic zones of the world. This ability is attributed to a well-developed mechanism of adaptation to a wide range of physical factors and their capacity to use multiple energy sources by regulating metabolism and only expressing proteins needed in certain environments and conditions. A well-equipped enzyme apparatus allows for the use of environments which are very poor in nutrients, e.g., building materials, plastics. The ability to synthesize a number of stress proteins, which protect mold fungi against extreme environmental conditions, also supports their expansiveness [17,18].

Biochar is a product of sustainable waste management. We currently know that it has found many applications in various areas of human life. Biochar is also an extremely innovative substance, due to its unique properties, which are constantly being discovered by scientists all over the world [19]. Sustainable development pertaining to new technologies is finding an increasingly wider scope of application in many areas of innovation. The protection of natural resources and the rational management of waste have led to the optimization of multiple technologies; hence the interest in biochar and its applications among both scientists and industry representatives. Despite numerous published research results and many reviews in journals published both in Polish and worldwide, there is still a lack of information about many of its properties. However, it should be emphasized that biochar has already been known about for a long time and thanks to the discovery of its properties, new possibilities of its application are also surfacing.

For many years, researchers have been undertaking work on the subject of preventive healthcare in the field of biological agent exposure. Progress is achieved through publishing valuable data that may inspire others in various fields and across multiple scientific disciplines. For this reason, based on the data related to the use of biochar, the authors of this study undertook interdisciplinary research related to the use of biochar as an ingredient in mortars in order to eliminate biologically harmful factors from building partitions and, consequently, from bioaerosols and room microclimates. The extremely challenging environments of historical buildings were selected as research sites, which, when exposed to climatic factors, biological corrosion, and the influence of direct and indirect factors, were subject to significant deterioration.

## 2. Materials and Methods

In many unsuccessful renovation or repair operations, it has been documented that neither clean lime mortars nor cement mortars prove effective in carrying out permanent repairs to walls loaded with moisture and salts [20]. Therefore, based on the information available in the relevant literature on renovation works and complementary mortars, biochar was selected as the basic component of cement mixtures being designed. This is an innovative raw material created in the process of biomass pyrolysis in the form of wood chips left over from mechanical processing of wood. The two most important properties of biochar are:low thermal conductivity;the ability to absorb water up to five times its own weight.

Such properties mean that biochar is an appropriate material for insulating and regulating humidity in historical buildings. Based on the analysis of the relevant literature, it can be concluded that one can make plastering mortars by combining lime with cement and biochar in proportions of up to 50% of the sand volume [21]. Plasters with the addition of biochar can:regulate room humidity within the range of 45–70% (optimal for human health);prevent the dehumidification of indoor air;prevent water condensation on the outer walls of the air, which leads to the formation of mold fungi;bind toxins and serve as an air purification method, as well as fungicide.

The production of biochar takes place in the process of biomass pyrolysis, which is heated to a temperature of 300–800 degrees Celsius in an oxygen-free environment. Biomass is boiled (roasted) in order to break down unnecessary products, such as lignin or cellulose, in order to create a hydrogen-rich fuel stream that can be burned or condensed to create energy. Biochar is a carbon-rich product which can be bound to hydrocarbons. The initial phase of the pyrolysis process is usually endothermic, meaning that it requires more energy than it produces. Ultimately, however, pyrolysis is exothermic, which means that it gives off more heat than required for commissioning (approx. 10% of final energy produced). In addition to the gas phase, a liquid phase is also obtained in the form of oil and meconium, and the remainder is coke breeze (up to 30% of the energy produced). About 5% of this energy is taken by the autothermal system. There is no liquid phase (tar and oils) during the production of biochar. This makes the pyrolysis process efficient, considering its carbon capture at the same time. Under the influence of heat (thermal decomposition), biomass is converted into biochar, when a constant, energetic stream of gases including carbon monoxide, methane and hydrogen are displaced from the biomass. The result is biochar, the mesh of the exoskeleton of carbon from the charred material, containing about 50% of the virgin carbon but only 30% of the remaining energy. The production process is controlled, and it allows one to freely shape the quality of biochar (from 65% to 93% of the carbon element content). The above-described technology is based patent No. 227338 belonging to the company FLUID S.A. from Sędziszów and relies on low-temperature (from 280 to 500 degrees Celsius) pyrolysis, as a result of which biochar is obtained (about 55% of the energy produced) and low-temperature hydrocarbon gases (about 45% of the energy produced) are directly produced [21]. Table 1 shows the most important properties of biochar.

The materials used for testing with various recipes were provided by FLUID S.A. from Sędziszów. The used materials differed from each other with regard to the carbon content in the biomass, being 70, 80 and 90%. The specified % in the mixture of materials used for the experiment was provided in a separate publication [20]. Preliminary studies showed good thermal insulation of the plaster and its positive effect on the microclimate of the rooms.

The wall surfaces as well as the air in the selected rooms of historical buildings were examined: the Bishop’s Castle (hydrophore room), and two churches: St. John the Baptist (vestibule, pillar) and the Holy Trinity in Janów Podlaski (crypt, altar), Lublin voivodeship, Poland. The research was carried out between 2018 and 2019. Microbiological swabs were collected from experimentally selected partitions (from the castle’s hydrophore room, the walls of the vestibule and pillars of St. John the Baptist Church, as well as from the altar and crypt of the Holy Trinity Church).

Microbiological analysis of biochar samples was based on the quantitative and qualitative determination of bacteria and mold fungi. Four standardized substrates were used for the tests to determine:The total number of colonies of mold fungi on an MEA substrate(MALT EXTRACT LAB-AGAR ™).The total number of fungal colonies on CYA(Czapek Yeast Autolysate Agar ™).The total number of colonies of mold fungi on a PDA substrate(POTATO DEXTROSE LAB-AGAR ™).The total number of microorganisms on a PCA substrate(PLATE COUNT LAB-AGAR ™).

Ready-made substrates by Graso (MEA, CYA, PDA, PCA), all with relevant quality certificates, were used, of which, in 2 replications, at 4 dilutions from 1 to 10^−3^, a total of 192 Petri dishes were used [22]. Qualitative and quantitative readings were made on the basis of the species composition of the microbial flora by using macroscopic and microscopic methods, as well as keys and taxonomic atlases, all expressed in CFU/g of biochar samples [6,7,8,9,10,11,22]. 

The types of microbiological substrates used for the analysis, their content and their research application are listed below [23,24,25]:Malt extract agar (MEA)—malt wort with chloramphenicol. Content: maltose extract (30.0 g), mycological peptone (5.0 g), agar (15.0 g), chloramphenicol (0.05 g), streptomycin (0.13 g). pH: 5.6 ± 0.2. Research application: isolation of mold fungi from the air and building partitions, storage and strain identification of mold fungi isolated from the air and building partitions, cultivation in a laboratory substrate (control).Czapek yeast autolysate (CYA) agar—yeast agar cap. Content: sucrose (30.0 g), Sodium nitrate (0.5 g), magnesium sulfate (3.0 g), potassium chloride (0.5 g), iron II sulfate (0.01 g), agar (13.0 g), dipotassium hydrogen phosphate (1.0 g). pH: 6.0 ± 0.2. Research application: breeding and isolation of microorganisms, identification of mold fungi isolated from building partitions, cultivation in a laboratory substrate (control).Potato-dextrose agar (PDA)—glucose-potato agar. Content: potato extract (4.0 g), dextrose (20.0 g), agar (15.0 g). pH: 5.6 ± 0.2. Research application: breeding and isolation of microorganisms, identification of mold fungi isolated from building partitions, cultivation in a laboratory substrate (control).Plate count agar (PCA)—agar with yeast extract, glucose and peptone. Content: enzymatic hydrolyzate of animal tissues (5.0 g), yeast extract (2.5 g), glucose (1.0 g), agar (15.0 g). pH: 7.0 ± 0.2. Research application: breeding, isolation and determination of the total number of microorganisms, cultivation in a laboratory substrate (control).

The breeding method consists of determining the number and type of fungi contained in the analyzed sample on the basis of their growth on solid substrates. The collected sample was placed on substrates. After 3–7 days of incubation at the temperature of 25–30 degrees Celsius using a laboratory incubator, the number of colonies was determined. After conversion, the number per unit area was given—CFU/g of biochar samples. In order to identify colonies of mold fungi, pure cultures were isolated from the grown colonies, establishing micro-cultures and, after incubation, macro- and microscopic (morphological) features of the isolated strains were assessed. They were then compared with the description in the relevant taxonomic keys in order to identify the genus and, if possible, the species. Breeding methods require up to 14 days and, in problematic cases, 21 days. Mycological analysis was carried out on the surface of building partitions. Samples of supplementary layers from building partitions were taken from an area of 100 cm^2^ by swabbing up to 50 cm^3^ of saline solution. The solution was serially diluted on a scale of 1:1000, and 1 cm^3^ of each dilution of the solution was plated on MEA substrate with chloramphenicol, CYA, PDA and PCA in two repetitions. The samples were incubated at 25–30 degrees Celsius for 3–7 days. After this time, the colonies were counted. The results were given as the number of CFU/100 cm^2^ in the area of the building partition.

An analysis of air contamination with mold fungi in the tested rooms was also carried out. Air sampling was carried out by means of a sedimentation method using MEA substrates. The sedimentation time of the collected air samples was 30 min. MEA substrates with chloramphenicol were used to grow mold fungi. Incubations were carried out at 25 degrees Celsius for 5 days. After this time, the colonies of microorganisms were counted and the results provided in CFU/m^3^ of air.

The swabs were taken in 3 stages—marking the collection site, collecting the material with a stick and placing it in a container with a saline solution of 0.9% NaCl. The number of colonies from the sedimentation plates to volume concentrations of CFU/m^3^ in air was converted using the following method described in the literature [26,27]:k_R_ = n/P × (5/t_S_)(1)
where:

k_R_—concentration of microbial contamination (CFU/m^3^),

n—number of colonies grown on the plate,

P—plate area (m^2^),

t_S_—tile opening time (sedimentation time in min, 30–60 min).

The isolated strains of mold fungi from air samples and building partitions of the studied historical buildings were grown and identified on malt extract agar (MEA), Czapek yeast autolysate (CYA) agar and potato-dextrose Agar (PDA) at a temperature of 25–30 degrees Celsius over 3–7 days. The process of developing the microscopic preparation consisted of placing a drop of saline with lactophenol on a glass slide, transferring a fragment of the mycelium with the tip of a preparation needle and covering it with a coverslip. Within each of the plates used, a total of nearly 500 preparations were created. The species identification was based on the available fungal atlases and diagnostic keys [7,11,23,24,25,26,27,28,29,30,31,32,33].

Microscopic analyses were performed using a Nikon Eclipse E-200 research microscope with fluorescence equipment and the SCA image analysis system. The NIS-Elements D photo processing and archiving software was used.

## 3. Results

On the basis of mycological analyses of wall surfaces and indoor air, the number and type of mold fungi were isolated and determined. A microscopic assessment of the types and species of mold fungi was also performed from the collected swabs. Next, the analysis of qualitative and quantitative composition of mold fungi identified in the air and on partitions was presented. The next stage of the research was the analysis of bioaerosol in rooms inside historical buildings. 

In the conducted microbiological analysis of the partitions, the following types and species of mold fungi were marked and identified in amounts ranging from 110 to uncountable (CFU/m^3^) on the MEA substrate at a dilution of 1: *Acremonium* sp., *Alternaria* sp., *Aspergilius* sp., *Clasosporium* sp., *Penicillium* sp., *Verticillium* sp. In the Bishop’s Castle in Janów Podlaski, on the other hand, amounts from 28 to uncountable (CFU/m^3^) were identified on the MEA substrate at the dilution of 1. In the Holy Trinity Church in Janów Podlaski, the amounts ranged from 0 to 258 [CFU/m^3^] on the MEA substrate at the dilution of 1. The data are compiled in Table 1. In the undertaken microbiological analysis of bioaerosols, the following genera and species of mold fungi were marked and identified: *Aspergilius* sp., *Clasosporium* sp., *Penicillium* sp., *Rhizopus* sp. In the Bishop’s Castle in Janów Podlaski, these species were identified at a quantity of 10–41 (CFU/m^3^); in the St John the Baptist church in Janów Podlaski, they were identified at a quantity of 4–48 [CFU/m^3^]; and in the Holy Trinity Church in Janów Podlaski, they were identified at a quantity of 3–24 [CFU/m^3^]. The data are summarized in Table 2.

After conducting the microbiological analysis:In the rooms of the Bishop’s Castle, in the first tested sample taken from the wall surface of the hydrophore room on the MEA substrate at a dilution from 1 to 10^−3^, the average number of colonies ranged from 66 to 110 CFU/100 cm^2^. Strains of *Aspergillius* sp. dominated. *Penicillium* sp. and *Verticillium* sp. were also isolated at a lower colony quantity. On the CYA and PDA substrates in the second sample from the hydrophore room at a dilution of 1, the plates were overgrown with *Penicillium* sp. Colonies, covering the entire surface. On plates with dilutions from 10^−1^ to 10^−3^, the average number of microorganisms noted was between 6 and 108 CFU/100 cm^2^. In the analysis of the tested air samples of the hydrophore room and banquet room on the MEA substrate, *Aspergillius* sp. and *Penicillium* sp. (from 10 to 41 CFU/m^3^) were isolated.At the St. John the Baptist church, in the tested sample acquired from the walls of the vestibule on the MEA substrate at a dilution of 1 in two plates, confluent growth was observed, which resulted in the lack of a quantitative reading. *Acremonium* sp., *Alternaria* sp., *Aspergilius* sp., *Clasosporium* sp. and *Penicillium* sp. were isolated from the remaining dilutions between 10^−1^ and 10^−3^ (average number of colonies from 1 to 36 CFU/100 cm^2^ at dilutions between 10^−1^ and 10^−2^). Strains of *Acremonium* sp. and *Alternaria* sp. dominated on the CYA substrate (from 5 to 153 CFU/100 cm^2^). The total number of microorganisms on the PDA substrate at dilutions between 10^−1^ and 10^−3^ ranged from 1 to 28, with strains of *Clasosporium* sp. being predominant. On the PCA substrate, the mean number of colonies ranged from 2 to 1672 CFU/100 cm^2^ at dilutions between 1 and 10^−3^. In the analyzed sample from the pillar on the MEA substrate at dilutions between 1 and 10^−1^, colonies of *Alternaria* sp. and *Penicillium* sp. were isolated. On the plates at a dilution between 10^−2^ and 10^−3^, no microorganisms were isolated. On the remaining substrates—CYA and PDA—single colonies of *Acremonium* sp., *Aspergilius* sp. and *Clasosporium* sp. were isolated. On the PCA substrate, confluent growth was observed on one of the plates, which resulted in the lack of a quantitative reading. The average number of colonies for the remaining substrates ranged from 1 to 52 CFU/100 cm^2^ at dilutions between 10^−1^ and 10^−3^. In the case of analyses conducted for the examined air samples in the rooms at St. John the Baptist church on the MEA substrate, *Aspergillius* sp., *Clasosporium* sp., *Penicillium* sp. and *Rhizopus* sp. (From 4 to 48 CFU/m^3^) were isolated.At the Holy Trinity church from the carried out microbiological analysis of the wall surfaces in the analyzed sample from the crypt on the MEA substrate, *Acremonium* sp., *Aspergilius* sp. and *Clasosporium* sp. (from 6 to 193 CFU/m^3^) were isolated. On plates at a dilution of 10^−3^, no microorganisms were isolated. On the CYA substrate at a dilution of 1, confluent growth was observed, which resulted in the lack of a quantitative reading, and *Clasosporium* sp. was isolated. On the PDA substrate, the number of colonies ranged from 1 to 372 CFU/100 cm^2^ at dilutions between 1 and 10^−3^. On the PCA substrate, the average number of colonies reached a level between 6 and 483 CFU/100 cm^2^ at dilutions between 1 and 10^−3^. In the case of the second crypt sample on the analyzed substrate, MEA, PDA-limited strains of *Clasosporium* sp. and *Penicillium* sp. were isolated. On plates with the CYA substrate, no microorganisms were isolated. On the PCA substrate, the average number of colonies reached 1 CFU/100 cm^2^. In the analyzed sample from the altar on the MEA, CYA, PDA substrates, single colonies of *Acremonium* sp. and *Cladosporium* sp. were isolated. The average number of colonies on the PCA substrate reached approximately 11 CFU/100 cm^2^. When it comes to the analysis of the examined air samples for the rooms at the Holy Trinity church on the MEA substrate, single units of *Aspergillius* sp., *Penicillium* sp. and *Cladosporium* sp. (from 3 to 24 CFU/m^3^) were isolated.

An analysis of the air in the rooms of the examined partitions of the historical buildings was also carried out. When it comes to the carried out microbiological analysis of the surface of walls at the Bishop’s Castle, to which a supplementary layer with the addition of biochar was applied, in the analyzed sample, namely the first hydrophore room sample on the MEA substrate at a dilution between 1 and 10^−3^, no microorganisms were isolated. On the remaining substrates—CYA and PDA—single units of *Penicillium* sp. were isolated. In the case of PCA, the total number of microbes reached 20 CFU/100 cm^2^. In the second hydrophore room sample on the MEA substrate, *Penicillium* sp. strains dominated (the average number of colonies ranged from uncountable to 20 CFU/100 cm^2^). The plates were overgrown with *Penicillium* sp. Colonies, covering the entire surface. When it comes to the two plates, confluent growth was observed, which resulted in the lack of a quantitative reading. The CYA and PDA substrates at dilutions between 1 and 10^−1^ were overgrown with *Aspergillius* sp. colonies. On the PCA substrate, the average number of colonies reached levels between 23 and 232 CFU/100 cm^2^ at dilutions between 10^−1^ and 10^−3^. When it comes to the carried out microbiological analysis of a wall at the St. John the Baptist church, where a supplementary layer with the addition of biochar was used, in the second analyzed sample from the vestibule on the MEA substrate at a dilution between 1 and 10^−3^, no microorganisms were isolated. On the remaining substrates of the analyzed samples (CYA, PDA), single units of *Penicillium* sp. and *Cladosporium* sp. were isolated.

Similarly, during the analysis of the examined air samples from rooms where a supplementary layer with the addition of biochar was used, when it comes to samples pertaining to the Bishop’s Castle, in the sample from the hydrophore room on the MEA substrate, single units of *Penicillium* sp. (from 4 to 9 CFU/m^3^) were isolated. When it comes to the rooms at St. John the Baptist church, among the species cultivated, the occurrence of species from the *Cladosporium* sp. and *Penicillium* sp. genus (from 76 to 84 CFU/m^3^), which typically occur in atmospheric air, was mostly determined. Epidemiological studies reported that mold fungi from *Aspergillius* sp., *Alternaria* sp., *Cladosporium* sp. and *Penicillium* sp. genera are most commonly found in buildings, and are the main source of allergens. The most common sources of allergies are substances present in the fungi *Cladosporium* sp. and *Alternaria* sp. Most patients are, however, hypersensitive to the allergens from several species of mold fungi at the same time [5,19,25,34,35,36,37,38].

Table 2 compiles values pertaining to the total value of isolated microorganisms from the partitions of the historical buildings analyzed before and after the application of supplementary mortar with the addition of biochar.

The concentration of microorganisms in the air of the tested rooms of historical buildings is below the norms proposed by Dutkiewicz and Gorny [2,16,21,26,27,28,29,39,40], described at the level of 200 CFU/m^3^ for mold fungi, which means that there is a low concentration of microorganisms and no measurable impact of the identified microorganisms on human health.

By comparing the period before and after the application of the supplementary mortar with the addition of biochar, when it comes to the Bishop’s Castle, the average number of microbes isolated from the examined partition before the application of supplementary mortar reached 61 CFU/100 cm^2^ on the MEA substrate and, after its application, this number decreased by almost 70%, reaching the level of 20 CFU/100 cm^2^. The average number of microbes isolated from the room’s air reached 23 CFU/m^3^ on the MEA substrate and, after the application, it decreased by 73% and reached 6 CFU/m^3^. At the St. John the Baptist church, in the sample from the examined partition on the MEA substrate at a dilution between 1 and 10^−3^, after the application of a mortar with the addition of biochar, no microorganisms were isolated. The average number of microbes isolated from the room’s air before the application of a mortar with the addition of biochar reached 44 CFU/m^3^ and, after the application, it increased slightly, reaching 80 CFU/m^3^ on the MEA substrate, which may be caused by the seasonal (fall-related) expansion of fungi spores in the air when the control experiment was being carried out. 

Table 3 compiles the values pertaining to the total number of microorganisms isolated from the air in the rooms of the examined historical buildings before and after the application of a supplementary mortar with the addition of biochar.

## 4. Discussion

On the basis of the available literature, it is possible to describe in detail the influence of particular genera and species of mold fungi on the human body. Therefore, identification is key in assessing exposure to harmful biological agents. The conducted research proved the presence of three types of mold fungi which are extremely dangerous to the human body: 

*Aspergillius* sp.—a genus of fungi from the Trichocomaceae family, occurring mainly in moist locations. Mold fungi of this genus reproduce by means of spores that form on the heads, giving them the shape of a sprinkler. Most species of aspergilloma (e.g., *A. fumigatus*) cause diseases in humans, such as skin or lung mycoses, or bronchial asthma. They also produce aflatoxins (e.g., *A. flavus*). Most mold fungi are mesophylls that grow at 18–35 degrees Celsius (they can also grow over a much wider temperature range). They show low sensitivity to temperatures below 0 degrees Celsius. Mold fungi of the above-mentioned type are sensitive to high temperatures. The commonly used pasteurization temperature destroys most of the vegetative forms of mold fungi. The spore structures can, however, remain viable. Some fungi, especially the genera *A. flavus* and *A. niger*, grow very well on highly salinated surfaces, which is a secondary effect of a long-term or strong exposure to moisture [2,4,5,6,8,9,10,11,14,16,17,18,19,20,21,22,25,26,27,28,35,39,40,41,42].

*Cladosporium* sp.—mold with the following taxonomic positions: Ascomycota, Pezizomycotina, Dothideomycetes, Dothideomycetidae, Capnodiales, and Davidiellaceae (teleomorph: Mycosphaerella). Hypersensitivity to this type of fungus can cause general weakness in the body, as well as lower immunity, which results in frequent colds and other infections. The most common symptoms of a Cladosporium allergy are sneezing, runny nose, asthma and fungal sinusitis. In particularly sensitive people, contact with Cladosporium may usually cause allergic reactions (e.g., atopic dermatitis). Colonies grow moderately fast; they are velvety, from mealy to wooly in texture, and grayish green to olive green in color.

The microscopic image of the colony most often presents branched hyphae, which are translucent or pigmented and smooth. Conidiophores are usually single, although not always, as they may occur in clusters. They are brown, straight, septated, may or may not be branched, and usually sporulating sympodially with a black-brown conidial scar. The spores are dry and cylindrical. They form branched chains that break easily. Most species do not grow at temperatures exceeding 35 degrees Celsius. Usually, this type of fungus is saprotrophic or phytopathogenic. Spores of this species are common in the air and occur on wood and wood-like materials. This fungus is parasitical to selected plants [1,2,3,7,8,9,10,11,13,14,15,16,17,18,21,22,23,24,25,26,27,28,29,30,31,34,39,40,41,42,43,44,45,46,47,48,49,50,51,52,53,54].

*Penicillium* sp.—mold with the following taxonomic positions: Ascomycota, Pezizomycotina, Eurotiomycetidae, Eurotiales, and Trichocomaceae (teleomorph: Eupenicillium, Talaromyces, Hamigera, Trichocoma). The symptoms of an allergy to this type of mold fungus may resemble an allergy to pollen (pollinosis), with a blocked nose being the main symptom. These ailments can both be periodic and chronic. Symptoms most often occur throughout the year, worsening in the autumn and summer periods, as they are directly related to the period of occurrence of mold spores in the air. Colonies are fast growing, flat, velvety, wooly or fluffy in texture, and initially white, then bluish-green, gray-green, olive green, yellow or pinkish in color. The underside of the colony is yellowish. The microscopic image shows hyphae 1.5–5 µm wide, which are single or branched, transparent, smooth or rough-sided. Conidiophores form a brush-like system, and at the top of the branches, metules with bottle-shaped phialides on them occur. Phialides can also grow at the top of an unbranched conidiophore. The spores are 2.5–5 μm in size, round, unicellular, transparent or greenish, and arranged in basal chains at the top of the phialides. Some species can form sclerotia. These fungi are widespread in nature, mostly found in the soil, on decaying plant debris and in the air [1,2,3,7,8,9,10,11,13,14,15,16,17,18,21,22,23,24,25,26,27,28,29,30,31,34,39,40,41,42,43,44,45,46,47,48,49,50,51,52,53,54].

The conducted analyses showed that, in terms of quality, the identified mold microflora of the studied historical buildings was essentially consistent, both when it comes to the castle and the religious buildings. There was no visible differentiation between the types and species of mold fungi present. The research results allow us to present seven types of mold fungi occurring in the studied historical buildings, i.e., *Acremonium* sp., *Alternaria* sp., *Aspergillius* sp., *Cladosporium* sp., *Penicillium* sp., *Rhizopus* sp. and *Verticillium* sp. Some of the defined fungi are classified as dangerous to the health of people with reduced immunity who are exposed to them. These include *Aspergillus fumigatus* and the allergenic *Aspergillus ustus* and *Altaernaria alternata*. Epidemiological studies show that fungi of the genera Alternaria, Cladosporium, Penicillium and Aspergillus, most commonly found in buildings, are also the most important allergens. The most common causes of allergies are substances present in the mycelium of *Alternaria alternata*. Most patients, however, are hypersensitive to the allergens of several species of fungi. Fungi from the genera *Penicillium* sp. and *Aspergillus* sp. can also be sensitizing. Commonly present in the environment, Aspergillus can cause internal organ diseases in humans by penetrating into the human body mainly through the respiratory tract. Small spores have the ability to survive in a wide range of temperatures and can reach all the way into the alveoli. Depending on the patient’s immune status, various forms of mycosis may occur, ranging from sinus allergies, through the development of mycelium in persistent cavities, all the way to invasive aspergillosis [1,2,3,7,8,9,10,11,13,14,15,16,17,18,21,22,23,24,25,26,27,28,29,30,31,34,39,40,41,42,43,44,45,46,47,48,49,50,51,52,53,54].

Inspections of mycological conditions of rooms inside selected historical buildings were also carried out after a supplementary layer had been applied. The aim of the study was to assess the mycological contamination of walls and air in the selected rooms of the examined historical buildings after the application of supplementary mortar with the addition of biochar. The subject of the research was the Bishop’s Castle and St. John the Baptist church in Janów Podlaski. Mycological studies were carried out four months after the completion of the renovation project. According to field-relevant literature, this was adequate time for the mycelium to develop, which approximately takes 3–6 weeks. Based on the undertaken analysis of laboratory cultures, a microscopic assessment of the types of mold fungi was completed and their total number was also determined. Including each of the substrates (MEA, CYA, PDA, PCA), with two repetitions and four dilutions (from 1 to 10^−3^), a total of 80 plates were used. Subsequently, the analysis of the qualitative and quantitative composition of mold fungi identified in the partitions and the air after the supplementary mixture was applied was presented.

Unfortunately, there are no similar studies available in the relevant literature, and therefore the discussion of the results is based only on the available literature data. In our work, we did not conduct studies on exposed persons, and rather only determined the in vitro potential impact of isolated mold fungi on health. Identifying microbiological indicators of damp and moldy buildings remains a challenge at the intersection between microbiology, construction science and public health. Adams et al. [50] tested 60 buildings in New York for moisture damage, and used three types of settled dust samples for microbiological analysis. The team examined the content of typically hydrophilic fungi and those that typically prefer humid environments. They also compared water-damaged houses with those that were undamaged by water for differences between indoor and outdoor fungal populations and differences in the presence or relative abundance of individual fungal taxa. According to the authors, hydrophilic fungi showed a significantly higher relative abundance in houses flooded with water, while mesophilic fungi, unexpectedly, had a much lower relative abundance in houses with visible mold. Mendell et al. [51] identified only three works in the literature related to dampness in buildings and its impact on the health of users. Two studies investigated the relationship between the experimentally measured moisture content in walls and the respiratory health of users in the UK. A dose-dependent severity of asthma exacerbations was noted at higher humidities. Another study investigated the relationship between wall humidity, as determined by an infrared camera, and South Korean atopic dermatitis in flooded buildings. Other studies conducted in the USA pertained to the collection of 12,026 air samples, 9619 indoor samples, and 2407 outdoor samples from 1717 buildings between 1996 and 1998. The samples were taken using an Andersen N6 instrument. The most commonly isolated strains belonged to the *Cladosporium* sp., *Penicillium* sp., non-sporulating fungi and *Aspergillus* sp. genera [52]. Similar results related to the isolated genera were obtained in our research. The genus *Penicillium* sp. was isolated, similarly to our research, from antique wood in the Arctic [53]. In contrast, Dedesko and Siegel [54] conducted a data review of drywall buildings. On the basis of the literature, they found that surface moisture, especially liquid water, is the main factor influencing microbiological changes, while attributing a lower importance to air and material moisture. Moreover, they describe that, after assessing surface moisture, a single critical moisture level to prevent fungal growth cannot be determined due to a number of factors, including differences in fungal types and/or species, temperature, and nutrient availability. Nevertheless, they recommend that such measurements be made to provide information on the development of fungi. This approach will capture changes in material surface moisture, which can provide insights into a range of conditions that may lead to fungal proliferation.

### The Limitations of the Study Design

There are many factors that may cause erroneous measurement results of microbiological purity, e.g., the wrong measurement method, inadequate location of measurement points, disturbance of the phenomenon by the experimenter, the emission of pollutants by the person performing the measurements, etc., as well as the country in which the experiment was carried out. On the one hand, this is justified because individual regions are characterized by a specific climate or microclimate. Hence, the standards recognized as correct in a given region of the world are justified for application there. On the other hand, comparing similar studies in different regions of the world may prove difficult. This issue may also apply to the microbiological media used, which, as we know, appear on the market very quickly, gradually replacing the classic ones that have been used for years. Identification plays a key role in microbiological research, which is also quite problematic. Unfortunately, it is not always possible to identify the tested sample at the species level, and therefore more care is required to specify the type of microorganisms in the results. Of course, the identification of species is extremely important due to a number of examples of toxin-causing and pathogenic microorganisms within one genus, and such a statement is possible only after identification at the species level.

The applied measurement procedure should minimize measurement errors and enable their evaluation. Despite the completely sterile protective clothing and extreme care employed during the measurements, the growth substrate can become contaminated, which may result in a false result. To determine the size of measurement errors, control plates should be attached to the series of plates with the substrate to indicate additional infections of the plates during manipulations related to the measurements performed and to check the sterility of the substrate and any contamination generated in the microbiological laboratory. Conclusions about the level of microbial contamination cannot be drawn from a single measurement. Due to the large number of factors disturbing the measurements and the large variability of the phenomenon, it is necessary to periodically perform a series of measurements and use statistical analysis to determine the measurement error.

## 5. Conclusions

The conducted analyses allow one to draw the following conclusions and indicate the following recommendations. The literature data and our own research indicate that fungi present in living and working environments may have a negative impact on human health. It is very important to follow all types of recommendations regarding the elimination of fungi from the construction industry. The problem of the occurrence of fungi in construction is still very current, and all activities aimed at the elimination of these microorganisms are also associated with widely understood and extremely important health prophylaxis.

The partitions of the above-mentioned buildings were infested with mold fungi. During the research in 2018–2019, swabs were collected in places infested with fungi (the castle hydrophore, walls of the vestibule and pillars of St. John the Baptist church, as well as the altar and crypt of the Holy Trinity Church). The results of the research reveal the presence of seven types of mold fungi, i.e., *Acremonium* sp., *Alternaria* sp., *Aspergillius* sp., *Cladosporium* sp., *Penicillium* sp., *Rhizopus* sp., *Verticillium* sp.

Microbiological analyses of the wall surfaces and air in selected rooms of the Bishop’s Castle, St. John the Baptist church, and the Holy Trinity church in Janów Podlaski, showed that in terms of quality, the identified mold microflora is qualitatively and quantitatively comparable between the castle and the religious buildings. The most frequently identified types of mold fungi were *Alternaria* sp., *Aspergillus* sp., *Cladosporium* sp., and *Penicillium* sp. The results of mycological tests carried out clearly indicate the potential properties of biochar inhibiting the growth of mold fungi. In the Bishop’s Castle, the average number of microorganisms isolated from the tested partition after application decreased by 70%. On the other hand, the average number of microorganisms isolated from the air in the room decreased by 73%. In the case of the walls of St. John the Baptist church, no microorganisms were isolated after the mixture with the addition of biochar was applied.

The accumulation of moisture in the tested rooms of historical buildings was stabilized. The applied supplementary layer contributed to the maintenance of room humidity in the optimal range of 60–70% for human health. Based on the analysis of the current state of knowledge and the conclusions drawn from the obtained research results—both theoretical and empirical—it is possible to formulate the main directions for the development of further works, which would concern pro-quality microbiological research.

## Figures and Tables

**Table 1 ijerph-19-12996-t001:** Identification of the material used in laboratory tests.

Property Name	Description
Trade name	Biochar
Product classification	According to regulation EC no. 1272/2008 [CLP/GHS], the substance is not classified as hazardous to health, or for the environment
Appearance	Physical state (20 degrees Celsius): solid, color: black
Smell	Lack
Density relative	Bulk density: in the range of 160–370 kg/m^3^
Water solubility	Practically insoluble
Temperature of self-ignition	Does not show self-ignition ability in the UN N.4 test
Reactivity	The substance is not reactive under normal conditions of use and storage
Chemical stability	The substance is used in recommended conditions of use and storage it is stable
Possibility of occurrence dangerous reactions	Under normal conditions of storage and use, hazardous reactions do not occur. Dusts can form explosive mixtures with air
Conditions to Avoid	Avoid fire sources
Incompatible materials	Strong oxidizers
Hazardous decomposition products	Under normal conditions of storage and use, the product does not decompose into hazardous materials; in a fire environment, dangerous carbon monoxide (CO) is released
Toxicity	The product has not been classified as hazardous to the environment
Persistence and degradability	The product is readily biodegradable
Root compactness coal	Above 70%
Chlorine, sulfur and mercury content	Trace amounts, less than 0.01%
Volatile matter content	Less than 17%
Ash content	Less than 6%

**Table 2 ijerph-19-12996-t002:** Mold fungi isolated on the partitions of the examined historical buildings before and after the application of supplementary mortar with the addition of biochar.

Sample No.	Collection Location	Substrate Type	Before the Application of Supplementary Mortar with the Addition of Biochar		After the Application of Supplementary Mortar with the Addition of Biochar	
			Total Number (CFU/100 cm^2^)	Types of Mold Fungi	Total Number (CFU/100 cm^2^)	Types of Mold Fungi
			Dilution		Dilution	
			10^−1^		10^−1^	
**Bishop’s Castle in Janów Podlaski**						
1	Hydrophore room 1	MEA	6.1	*Aspergillius* sp. *Penicillium* sp. *Verticillium* sp.	0	*Aspergillius* sp. *Penicillium* sp.
		CYA	10.8		0	
		PDA	21.5		0	
		PCA	20.0		0.2	
2	Hydrophore room 2	MEA	16.5		unc.	
		CYA	10.8		unc.	
		PDA	21.5		unc.	
		PCA	20.0		32.3	
**St. John the Baptist church in Janów Podlaski**						
3	Vestibule 1	MEA	3.6	*Acremonium* sp. *Alternaria* sp. *Aspergilius* sp. *Clasosporium* sp. *Penicillium* sp.	0	*Cladosporium* sp. *Pencilium* sp.
		CYA	2.5		0	
		PDA	2.8		0.1	
		PCA	25.7		0	
4	Vestibule 2	MEA	3.1		0	
		CYA	3.6		0	
		PDA	6.1		0	
		PCA	25.2		0	

unc.—uncountable.

**Table 3 ijerph-19-12996-t003:** Mold fungi isolated from the air in the rooms of the examined historical buildings before and after the application of supplementary mortar with the addition of biochar.

Sample No	Collection Location	Before the Application of Supplementary Mortar with the Addition of Biochar		After the Application of Supplementary Mortar with the Addition of Biochar	
		Total Number (CFU/m^3^)	Types of Mold Fungi	Total Number (CFU/m^3^)	Types of Mold Fungi
**Bishop’s Castle in Janów Podlaski**					
1	Hydrophore room 1	10	*Aspergillius* sp. *Penicillium* sp.	4	*Penicillium* sp.
2	Hydrophore room 2	24		5	
**St. John the Baptist church in Janów Podlaski**					
3	Vestibule 1	40	*Aspergilius* sp. *Clasosporium* sp. *Penicillium* sp. *Rhizopus* sp.	76	*Aspergilius* sp. *Clasosporium* sp. *Penicillium* sp.

## Data Availability

The data that support the findings of this study are available in Bialystok University of Technology and Pope John Paul II State School of Higher Education in Biala Podlaska.

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
