# Peer review of "The Impact of Biochar Used in Repairs to Historical Buildings on Public Health"

_ijerph, 2022, doi:10.3390/ijerph192012996_

Round 1

Reviewer 1 Report

The article presents important issues related to the mycology of construction, and in particular deals with the problem of the occurrence of mold fungi in historic buildings. This is an important topic because mold fungi can pose a health risk to people staying in buildings, but also to the buildings themselves (finishing materials), causing serious material losses. Researchers identified the presence of typical microorganisms that are isolated on mineral materials. In my opinion, the publication can be published after supplementing and clarifying some important issues.

Introduction:

Apart from VOCs, an important factor responsible for the occurrence of the so-called The Sick Building Syndrome is (1 → 3) β-glucan. This is worth mentioning in the literature review.

Materials and methods:

The sedimentation method in the assessment of mycological air pollution is not recommended due to inaccurate estimation of small particles. It would be better to use volumetric methods, e.g. the impact method, for greater accuracy of the test.

Results:

Table 2: The microorganisms isolated on the surface of Pillar 1 and Pillar 2 are not listed.

The research lacks measurements and analysis of one of the very important factors of mold growth in buildings - the measurement of humidity. If the moisture content of the materials is too high, the anti-mold action will not be effective. Looking at the results in Table 4, and more specifically at the condition of hydrophore 1 and 2 fungus, you can see a serious difference between the rooms. Due to the fact that these rooms are located in the same building and the renovation works were probably carried out at the same time, it can be seen that for some reason the biochar used did not work in the "hydrophore 2" room. Presumably, the humidity of building partitions was of great importance here. In the conclusions, the authors of the research provide a humidity range of 60-70% (relative air), but there is no information about the humidity of building partitions. The methodology also did not provide information on the measurement of the humidity mentioned in the conclusions, either before the work or after the use of biochar.

The authors do not provide anything about the determined indor/outdoor (I/O) ratio, which would be helpful especially for the interpretation of the results of microbiological air quality tests in St. John the Baptist chuch (there were less microorganisms in the air before the use of biochar than after the use of this component). Thanks to the I/O ratio, it can be determined whether the greater number of CFU/cm3 in the air in the building after the application of the biochar is due to the higher level of outdoor air pollution. If the I/O were above 1, it would mean that the fungus comes from inside, and this would result in either a lack of effectiveness of the biochar used or too high humidity. On the surface of building partitions in the "vestibule 1" room, the measurement result of microbial contamination is very low, so the greater amount of microorganisms in the air cannot be the cause of the development of fungi in the building. The reason for this may also be the very low accuracy of the sedimentation method.

The authors of the study did not identify the species of fungi. They only identified the type of fungi. The use of 4 types of culture media and the use of microbiological identification keys, for example the items 7 and 27 mentioned in the literature, definitely allow for a wider identification - up to the species. Identification at the species level would be important from the point of view of the described harmfulness of fungi to health and also indicate whether the species composition of fungi changes or remains unchanged before and after the use of biochar. Perhaps some species will be more sensitive to biochar and others will not.

Discission:

Please write the listed species of fungi in italics (line: 343, 344, 349).

Line: 351 - To be connected to line 352. Line 352 "Cladosporium…" should be started with a new line.

Line: 394 should be Alternaria alternata.

Line: 399 sentence "Fungi from genera Penicillium sp. and Aspergillus sp." is unclear.

Line: 400, please specify which types belong to Aspergillus or improve on Aspergillus sp.

Line: 426 please correct the reference literature and remove the year of publication here.

Line: 434 similar to above.

Line: 445 Cladosporium sp., Penicillium sp. should be written in italics.

Line: 447 similar to above.

Line: 448 Dedesko and Siegel (2015) should be Dedesko and Siegel [50].

The discussion of the results is mainly based on the analysis of the harmfulness of mold fungi. On the other hand, there is no reference to the assessment of the effectiveness of the applied biochar in the elimination of microbiological contamination and the comparison of its effectiveness in relation to other substances commonly used in construction.

Therefore, the conclusions should also be revised. Conclusions from the conducted research should not contain references to the literature, should constitute a summary of the work and indicate its most important results.

Author Response

Review Response 1

Title of the article:

The impact of biochar used in repairs of historical buildings on public health

Comment 1: Apart from VOCs, an important factor responsible for the occurrence of the so-called The Sick Building Syndrome is (1 → 3) β-glucan. This is worth mentioning in the literature review.

Response 1: Following the guidelines contained in the review, the authors introduced appropriate corrections and additions.

Comment 2: The sedimentation method in the assessment of mycological air pollution is not recommended due to inaccurate estimation of small particles. It would be better to use volumetric methods, e.g. the impact method, for greater accuracy of the test.

Response 2: Sharing the Reviewer's remark fully on the methodology used, it will be extended to include volumetric methods in subsequent publications.

Comment 3: Table 2: The microorganisms isolated on the surface of Pillar 1 and Pillar 2 are not listed.

Response 3: The microorganisms isolated on the surfaces of pillar 1 and pillar 2, after correction, were listed in the line 257.

Comment 4: The research lacks measurements and analysis of one of the very important factors of mold growth in buildings - the measurement of humidity. If the moisture content of the materials is too high, the anti-mold action will not be effective. Looking at the results in Table 4, and more specifically at the condition of hydrophore 1 and 2 fungus, you can see a serious difference between the rooms. Due to the fact that these rooms are located in the same building and the renovation works were probably carried out at the same time, it can be seen that for some reason the biochar used did not work in the "hydrophore 2" room. Presumably, the humidity of building partitions was of great importance here. In the conclusions, the authors of the research provide a humidity range of 60-70% (relative air), but there is no information about the humidity of building partitions. The methodology also did not provide information on the measurement of the humidity mentioned in the conclusions, either before the work or after the use of biochar.

            The authors do not provide anything about the determined indor/outdoor (I/O) ratio, which would be helpful especially for the interpretation of the results of microbiological air quality tests in St. John the Baptist chuch (there were less microorganisms in the air before the use of biochar than after the use of this component). Thanks to the I/O ratio, it can be determined whether the greater number of CFU/cm3 in the air in the building after the application of the biochar is due to the higher level of outdoor air pollution. If the I/O were above 1, it would mean that the fungus comes from inside, and this would result in either a lack of effectiveness of the biochar used or too high humidity. On the surface of building partitions in the "vestibule 1" room, the measurement result of microbial contamination is very low, so the greater amount of microorganisms in the air cannot be the cause of the development of fungi in the building. The reason for this may also be the very low accuracy of the sedimentation method.

            The authors of the study did not identify the species of fungi. They only identified the type of fungi. The use of 4 types of culture media and the use of microbiological identification keys, for example the items 7 and 27 mentioned in the literature, definitely allow for a wider identification - up to the species. Identification at the species level would be important from the point of view of the described harmfulness of fungi to health and also indicate whether the species composition of fungi changes or remains unchanged before and after the use of biochar. Perhaps some species will be more sensitive to biochar and others will not.

Response 4: The authors fully agree with the comments of the Reviewer and have introduced appropriate corrections to the text, if possible, in order to improve its substantive value and scientific precision.

Comment 5: Editing notes listed on lines 343-488.

Response 5: The listed editorial notes in lines 343-488 have been corrected,

Comment 6: The discussion of the results is mainly based on the analysis of the harmfulness of mold fungi. On the other hand, there is no reference to the assessment of the effectiveness of the applied biochar in the elimination of microbiological contamination and the comparison of its effectiveness in relation to other substances commonly used in construction.

            Therefore, the conclusions should also be revised. Conclusions from the conducted research should not contain references to the literature, should constitute a summary of the work and indicate its most important results.

Comment 6:

Thank you for your valuable comment. Sources were removed from Conclusions as recommended by the reviewer

Additional information/remarks:

The authors read the content of the review with great interest and thank the Reviewer for the time and effort put into its preparation, and above all for the undoubtedly constructive comments. For the sake of the high-quality content of the publication, the corrections suggested in the review were taken into account. In the opinion of the authors, the comments submitted in the review were justified.

Authors

Reviewer 2 Report

1. Short description of the study

The aim of this manuscript is the study of the effect of biochar on certain microorganisms and, accordingly, the interest of the use as a complement to the material used to repair historical buildings. The hypothesis of this work postulates that the proliferation over time of microorganisms that negatively affects human health could be controlled/reduced with the use of supplementary layers of biochar in the partitions of the walls. Microbiological analyses of the surface of the walls and air were carried out before and after the application of biochar layers.

2. Manuscript´s strengths

The use of biochar as a component of building materials is a topic of great interest and popularity. This study presents the results of experiments that combine biochar and microorganisms within an original context in which the proliferation of microorganisms has a special relevance. There is an interest to improve the characteristics of conventional materials through the use of sustainable alternatives.

3. Manuscript´s weaknesses

In general terms, shortcomings are observed in the experimental design and the description of the methodology that should be modified before publishing the work.

Regarding the experimental design, there is not a control treatment to isolate the effect of biochar in the material used to repair the walls. Considering this, it can not be affirmed that it is the biochar and not the mixture of components (or the simple sealing of the walls) that reduces the concentration of microorganisms.

From my point of view, a work that focuses on the use of biochar should correctly describe the material used, since biochar is not a unique material and the characteristics vary a lot depending on the feedstock and the operating conditions in which it is obtained. In addition, although there are not previous studies on the use of biochar for the purpose of this work, information on biochar used in construction materials should be included in the introduction section, considering that the authors have experience in that subject. Furthermore, it is possible to find a lot of information on the interaction between biochar and soil microbiology, which could help to interpret the results obtained in this work.

Shortcomings have also been detected in the methodology used and the presentation of results related to the microbiology. These aspects are detailed in the following point of the report.

4. Specific comments

As a recommendation, the following characteristics of biochar should be detailed:

-        Feedstock characteristics

-        Pyrolysis operating conditions (Temperature, pressure and residence time of the carrier gas within the reactor)

-        Physicochemical characteristics of the obtained biochar (please pay attention to L97 in which a risky claim has been made)

-        Type of mechanical processing and granulometric distribution (please consider that the nature of feedstock and the distribution of the granulometry of biochar is important to evaluate the concentration of particles PM10 contained in the biochar and susceptible to be emitted into the environment).

-        Specific % in the mixture of materials used for the experiment

Regarding the methodology adopted to study the microorganisms:

-        Authors selected MEA, CYA, PDA, PCA, neither of which is suitable for counting fungi, they are usually used for isolation and cultivation; PCA is a counting agar for bacteria and general counts of fungi species, it is not suitable for the purpose of this work. The authors should have use Dichloran Rose Bengal Chloramphenicol (DRBC) agar, which allows a reliable fungal count. That is the reason why the authors have so many countless results.

-        Fungi species are cultivated at a range of temperature from 25 °C to 30 °C; according to ISO 7954:1987, it should be a constant 25 °C since there may be fungi that do not correctly develop at 30 °C.

-        The authors have used superficial swabbing in 100 cm2. Perhaps using superficial scrapping would capture a greater number of fungi species.

-        A very important question: there is not a control treatment and there is not a microbiological analysis of the material before to be used to repair the walls.

-        There is a standardized methodology for environmental sampling, Index of Microbial Air Contamination (IMA), and ISO 14698. The methodology adopted in this study does not match to either of these.

L125: Microbiological analyses of biochar samples. The procedure of the use of biochar has not been previously explained.

Regarding the reported results:

-        The tables are not well expressed. Firstly, not all the dilutions are presented, only a final total result is presented, making the corresponding operations with the different dilutions.

-        Where is the detection limit? It can not be found results below detection limit, which will be 50 CFU/100 cm2, according to what the authors have proposed.

-        Results presented in tables are inconsistent. It is not possible to obtain counts of 0.x or 0.0.x or 0.00.x in a Petri dish in the different dilutions and less with only 2 repetitions, the result should be at least 1. Considering that the final result will always be above of 50 CFU/cm2. For example, Table 1, sample 3, hydrophore room 1, PCA: in dilution 1 counts of 1672 CFU/cm2 to 25.7 CFU/cm2 in dilution -1, that is inconsistent, a good count should be 167 CFU/cm2 in dilution -1. Or unc. in dilutions 1 to counts of 3 to 25 in dilutions -1 in all tables. These results are repeated along the tables.

-        Results of environmental samples: the authors present the results in CFU/100 cm3, this is carried out using a forced air sampler type SMPL´AIR (bioMerieux), in which a certain volume of air passes through the Petri dish. In this case it can not be expressed in this way, please consult the regulations for ambient air samples.

From a microbiological point of view, this experiment is not correctly designed and executed. It has not sense to present the counts of different dilutions, it should be presented a final result. They should have used DRBC medium for a reliable fungal count, and consider what detection limit they will have according to the sampling carried out, in this work 50 CFU/cm2.

Author Response

Review Response 2

Title of the article:

The impact of biochar used in repairs of historical buildings on public health

Comment 1: As a recommendation, the following characteristics of biochar should be detailed.

Response 1: Based on the submitted comments of the Reviewer, the features of biochar have been listed in the introduction.

Comment 2: Pyrolysis operating conditions (Temperature, pressure and residence time of the carrier gas within the reactor)

Response 2: : Thank you for your very valuable remark. Biochar was produced according to Fluid SA company technology. The process consisted in thermal refining of plant biomass and other post-production biomass residues through their autothermal roasting at the temperature 260°C in reduction atmosphere and without the use of additional energy, catalysts, and chemical additives. Carbonising products consisted mainly of biochar (from 65% to 70% of energy compared to the energy of the feed) and process gases (from 20% to 35% of energy compared to the energy of the feed). During the process, a significant increase of the carbon element (C) in relation to the biomass of the feed was recorded (1.5 to 2.0 times) as well as an increase of energy density, on average by 4.0 times, reduction of the amount of hydrogen (H), on average by 2.5 times, and reduction of the amount of oxygen (O2), on average by 3.0 times.

Comment 3: Physicochemical characteristics of the obtained biochar (please pay attention to L97 in which a risky claim has been made)

Response 3: : Thank you for your very valuable remark. The information comes from the source: Gładki, J. Biowęgiel szansą dla zrównoważonego rozwoju. Printing House Apla,: Sedziszow, Poland, 2017.

Comment 4: Type of mechanical processing and granulometric distribution (please consider that the nature of feedstock and the distribution of the granulometry of biochar is important to evaluate the concentration of particles PM10 contained in the biochar and susceptible to be emitted into the environment).

Response 4: : Thank you for your very valuable remark. Such studies have not been conducted.

Comment 5: Specific % in the mixture of materials used for the experiment.

Response 5:

Thank you for your very valuable remark. The information comes from the source: Tokarski, D.; Ickiewicz, I. Naprawy zabytkowych murów warstwami uzupełniającymi z dodatkiem biowęgla. Publishing House of the Bialystok University of Technology: Bialystok, Poland, 2021.

Comment 6: Regarding the methodology adopted to study the microorganisms: Authors selected MEA, CYA, PDA, PCA, neither of which is suitable for counting fungi, they are usually used for isolation and cultivation; PCA is a counting agar for bacteria and general counts of fungi species, it is not suitable for the purpose of this work. The authors should have use Dichloran Rose Bengal Chloramphenicol (DRBC) agar, which allows a reliable fungal count. That is the reason why the authors have so many countless results.

Response 6: Thank you for your very valuable remark. In this experiment, we used the substrate for isolation and cultivation, due to the fact that we did not know the substrate suggested by the reviewer. We used media that had been used by other authors in similar studies.

Of course, we will apply this remark in our future research.

Comment 7: Fungi species are cultivated at a range of temperature from 25 °C to 30 °C; according to ISO 7954:1987, it should be a constant 25 °C since there may be fungi that do not correctly develop at 30 °C.

Response 7: Thank you for your very valuable remark. The temperature was given in the range of 25-30 ° C. The temperature (25 ° C) was used on the MEA and CYA substrates, while the temperature (30 ° C) was used on the PDA substrate.

Comment 8: The authors have used superficial swabbing in 100 cm2. Perhaps using superficial scrapping would capture a greater number of fungi species.

Response 8: Thank you for your valuable comment. Of course, we will apply this remark in our future research. attention

Comment 9: A very important question: there is not a control treatment and there is not a microbiological analysis of the material before to be used to repair the walls.

Response 9: Thank you for your valuable comment. We have based on the literature data related to the beneficial effects of biochar on the environment. Such studies have not been conducted.

Comment 10: There is a standardized methodology for environmental sampling, Index of Microbial Air Contamination (IMA), and ISO 14698. The methodology adopted in this study does not match to either of these.

Response 10: Thank you for your valuable attention. Our research was an innovative experiment. We relied on the method used in Poland:

  • PN-EN ISO 7218Microbiology of Food and Animal Feeding Stuffs—General Rules for Microbiological Examination; Polish Committee for Standardization: Warsaw, Poland, 2013.
  • Ramirez, C. Manual and Atlas of the Penicillia; Elsevier Biomedical Press: Amsterdam, The Netherlands, 1982.
  • Baran, E. Zarys Mikologii Lekarskiej; Volumed: Wrocław, Poland, 1998.
  • Larone, D.H. Medically Important Fungi. A Guide to Identification, 5th ed.; ASM Press: Washington, DC, USA, 2011.
  • Kwaśna, H.; Chełkowski, J.; Zajkowski, P. Grzyby; Instytut Botaniki PAN: Kraków, Poland, 1991; Volume XXII .
  • Krzyściak, P.; Skóra, M.; Macura, A.B. Atlas Grzybów Chorobotwórczych Człowieka; MedPharm: Wrocław, Poland, 2011.
  • Samson, R.A.; Hoekstra, E.S.; Frisvad, J.C.; Filtenborg, O. Introduction to Food- and Airborne Fungi,6th ed.;Centraalbureau voor Schimmelcultures: Utrecht, The Netherlands, 2002.

Comment 11: Microbiological analyses of biochar samples. The procedure of the use of biochar has not been previously explained.

Response 11: Thank you for your valuable attention. We have completed the above data.

Comment 12: Regarding the reported results: The tables are not well expressed. Firstly, not all the dilutions are presented, only a final total result is presented, making the corresponding operations with the different dilutions.

Response 12: Thank you for your valuable comment. Due to the large number of analyzes and the limited number of pages in the journal, we decided to provide only the final total score.

Comment 13: Where is the detection limit? It can not be found results below detection limit, which will be 50 CFU/100 cm2, according to what the authors have proposed.

Response 13: Thank you for your valuable remark. We adopted the lower limit of detection as 50 CFU / 100 cm2

Comment 14: Results presented in tables are inconsistent. It is not possible to obtain counts of 0.x or 0.0.x or 0.00.x in a Petri dish in the different dilutions and less with only 2 repetitions, the result should be at least 1. Considering that the final result will always be above of 50 CFU/cm2. For example, Table 1, sample 3, hydrophore room 1, PCA: in dilution 1 counts of 1672 CFU/cm2 to 25.7 CFU/cm2 in dilution -1, that is inconsistent, a good count should be 167 CFU/cm2 in dilution -1. Or unc. in dilutions 1 to counts of 3 to 25 in dilutions -1 in all tables. These results are repeated along the tables.

Response 14: Thank you for your valuable comment. The actual result is presented in the table. We will apply this valuable consideration in future studies

Comment 15: Results of environmental samples: the authors present the results in CFU/100 cm3, this is carried out using a forced air sampler type SMPL´AIR (bioMerieux), in which a certain volume of air passes through the Petri dish. In this case it can not be expressed in this way, please consult the regulations for ambient air samples.

Response 15: Thank you for your very valuable comment. Air sampling was carried out by the classical sedimentation method with the use of MEA media, described in the literature. The sedimentation time of the collected air samples was 30 minutes. MEA substrates with chloramphenicol were used to grow mold fungi. Incubations were carried out at 25 degrees Celsius for 5 days. After this time, the colonies of microorganisms were counted, the results are given in cfu / m3 of air. In particular, the knowledge contained in the sources was used:

  • B.: Mikrobiologia powietrza. Wydawnictwo Politechniki Warszawskiej, Warszawa 1992, 19-20
  • PN – 89 – 04111/03 Ochrona czystości powietrza. Badania mikrobiologiczne. Oznaczanie liczby grzybów mikroskopowych w powietrzu atmosferycznym (imisja) przy pobieraniu próbek metodą aspiracyjną i sedymentacyjną.
  • Nabrdalik M.: Grzyby strzępkowe w obiektach budowlanych. Ecological Chemistry and Engineerings, 2007, 14, 489-496.

Comment 16: From a microbiological point of view, this experiment is not correctly designed and executed. It has not sense to present the counts of different dilutions, it should be presented a final result. They should have used DRBC medium for a reliable fungal count, and consider what detection limit they will have according to the sampling carried out, in this work 50 CFU/cm2.

Response 16:

Thank you for your valuable remark. We will apply all your valuable comments in the next research

Additional information/remarks:

The authors thank the Reviewer for a thorough assessment of the content of the submitted text of the publication, as well as the included critical comments. They also appreciate the commitment of the Reviewer in striving to remove substantive and editorial defects. In the opinion of the authors, the comments submitted in the review were justified. For the sake of high substantive quality of the publication, the suggested corrections have been taken into account.

Authors

Round 2

Reviewer 1 Report

The Authors of the study referred to many comments, although I still lack a good justification for the lack of moisture testing of building partitions, which is extremely important when designing this type of research and assessing the effectiveness of fungicides. However, I hope that the Authors of the publication, as stated in response to the review, will design further studies, taking into account the guidelines contained in the review.

Author Response

Review Response 1 (Round 2)

Title of the article:

The impact of biochar used in repairs of historical buildings on public health

Comment 1: The Authors of the study referred to many comments, although I still lack a good justification for the lack of moisture testing of building partitions, which is extremely important when designing this type of research and assessing the effectiveness of fungicides. However, I hope that the Authors of the publication, as stated in response to the review, will design further studies, taking into account the guidelines contained in the review.

Response 1: Thank you for your very valuable remark. Of course, testing the moisture content of materials is the most difficult to measure, and is also a key parameter to assess the results obtained. The methodology used and the results of measuring the moisture content in building partitions are described in the source: Tokarski, D.; Ickiewicz I. Naprawy zabytkowych warstwami warstwami uzupełniającymi z biowęglami. Wydawnictwo Politechniki Białostockiej: Białystok, Polska, 2021.

Additional information/remarks:

The authors thank the Reviewer for a thorough assessment of the content of the submitted text of the publication, as well as the included critical comments. They also appreciate the commitment of the Reviewer in striving to remove substantive and editorial defects. In the opinion of the authors, the comments submitted in the review were justified. For the sake of high substantive quality of the publication, the suggested corrections have been taken into account.

Authors

Reviewer 2 Report

Dear authors,

the deficiencies detected in the work were detailed in the previous review report:

- Experimental design (lack of control treatment)...

- Methodology (lack of description of the characteristics of the biochar used, restoration and sampling process, use of adequate culture media...)

- Presentation of results.

These deficiencies require new analysis and cannot be resolved in a few days. In addition, the authors have only added a general paragraph in the introduction (and not in methodology) that does not adequately decribe the characteristics of the biochar used (considering the characteristics detailed in the previous report). 

I can not support the publication of this manuscript in the present form.

Author Response

Review Response 2 (Round 2)

Title of the article:

The impact of biochar used in repairs of historical buildings on public health

Comment 1:

1) Experimental design (lack of control treatment).

2) Methodology (lack of description of the characteristics of the biochar used, restoration and sampling process, use of adequate culture media).

3) Presentation of results.

These deficiencies require new analysis and cannot be resolved in a few days. In addition, the authors have only added a general paragraph in the introduction (and not in methodology) that does not adequately decribe the characteristics of the biochar used (considering the characteristics detailed in the previous report).

Response 1:

Thank you for the valuable attention of the Reviewer. Of course, microbiological tests are the most difficult to measure, and are also a key parameter to evaluate the results obtained. The methodology used does not fully satisfy us. The main reason was the lack of access to specialized research equipment.

The main premise that we followed when selecting the composition of the tested mortars were the guidelines provided by the Investor. Our task was to optimally select the proportions (recipes) to obtain the appropriate technical parameters and achieve the desired effect. Moreover, when analyzing the literature on the subject, we found records that damage to walls is often caused by too tight and too strong mortars in relation to porous bricks. Large and deep defects in the face of the wall thread, and the cement joint remains intact around it [1]. So it seemed to me the right step to use biochar as a composite with cement, having a porous structure.

In the first stage of works, the historic joint was chained 15 cm deep into the wall. In the second stage, the furrows were thoroughly cleaned, and in the next stage complementing them with composite with biochar. Mycological studies were carried out four months after the completion of the works. According to the literature from a given range, this time was adequate to the period of mycelium development, which is approximately 3-6 weeks [2].

The research was carried out in 2018-2019, which makes it difficult to conduct a new analysis. All known properties of the biochar used have been precisely described in the article - the above-mentioned information has been transferred to the methodology.

References

  1. Koprowicz R. (2016). Zasady doboru zapraw do prac renowacyjnych przy zabytkowych murach ceglanych i kamiennych. Technologia materiałów według wytycznych konserwatorskich i norm budowlanych. Inżynier budownictwa, nr 02/2016.
  2. Gutarowska B. (2010). Grzyby strzępkowe zasiedlające materiały budowlane wzrost oraz produkcja mikotoksyn i alergenów. Zeszyty Naukowe, Politechnika Łódzka, nr 1074.

Additional information/remarks:

The authors read the content of the review with great interest and thank the Reviewer for the time and effort put into its preparation, and above all for the undoubtedly constructive comments. For the sake of the high-quality content of the publication, the corrections suggested in the review were taken into account. In the opinion of the authors, the comments submitted in the review were justified.

Authors
